

# A systematic analysis of the science of sandboxing

Michael Maass[1], Adam Sales[2], Benjamin Chung[1] and Joshua Sunshine[1]

[1] Institute for Software Research, School of Computer Science, Carnegie Mellon University, Pittsburgh, PA, United States
[2] Statistics Department, Carnegie Mellon University, Pittsburgh, PA, United States

## ABSTRACT

Sandboxes are increasingly important building materials for secure software systems. In recognition of their potential to improve the security posture of many systems at various points in the development lifecycle, researchers have spent the last several decades developing, improving, and evaluating sandboxing techniques. What has been done in this space? Where are the barriers to advancement? What are the gaps in these efforts? We systematically analyze a decade of sandbox research from five top-tier security and systems conferences using qualitative content analysis, statistical clustering, and graph-based metrics to answer these questions and more. We find that the term "sandbox" currently has no widely accepted or acceptable definition. We use our broad scope to propose the first concise and comprehensive definition for "sandbox" that consistently encompasses research sandboxes. We learn that the sandboxing landscape covers a range of deployment options and policy enforcement techniques collectively capable of defending diverse sets of components while mitigating a wide range of vulnerabilities. Researchers consistently make security, performance, and applicability claims about their sandboxes and tend to narrowly define the claims to ensure they can be evaluated. Those claims are validated using multi-faceted strategies spanning proof, analytical analysis, benchmark suites, case studies, and argumentation. However, we find two cases for improvement: (1) the arguments researchers present are often *ad hoc* and (2) sandbox usability is mostly uncharted territory. We propose ways to structure arguments to ensure they fully support their corresponding claims and suggest lightweight means of evaluating sandbox usability.

## INTRODUCTION

Sandboxes can be found where software components must be used but cannot currently be verified or trusted. Sandboxed components are often feared to be either malicious or vulnerable to attack. For example, popular browser engines (e.g., Google Chrome and Internet Explorer), productivity software (e.g., Microsoft Word and Adobe Reader), and operating system kernels (e.g., Windows 8) have all been sandboxed to varying degrees. Virtual machines, which run software on simulated hardware separate from the rest of the

Corresponding author
Michael Maass, mmaass@andrew.cmu.edu

host system, are commonly used in malware analysis to contain malicious computations (e.g., Cuckoo) and sandboxes are used in mobile ecosystems (e.g., Android) to limit the havoc malicious applications can wreak on a device.

Sandboxes provide some salvation in a world full of complex components: instead of vetting intractably enigmatic software systems, sandbox them and vet the relatively simple sandbox. What else do sandboxes in mainstream use have in common? They are largely dependent on relatively easily composed coarse-grained operating system features, they are essentially transparent to the users of sandboxed components, and they make little use of decades worth of research produced within the domain of software security sandboxing.

Researchers have spent the last several decades building sandboxes capable of containing computations ranging from fully featured desktop applications to subsets of nearly every kind of application in existence, from third party libraries in Java programs to ads on web sites. Sandboxes have been built to stop memory corruption exploits, ensure control- and data-flow integrity, enforce information flow constraints, introduce diversity where monocultures previously existed, and much more. What more can the community do to bring value?

In this paper, we use multidisciplinary techniques from software engineering, statistics, the social sciences, and graph analysis to systematically analyze the sandboxing landscape as it is reflected by five top-tier security and systems conferences. We aim to answer questions about what sandboxes can already do, how they do it, what it takes to use them, what claims sandbox inventors make about their creations, and how those claims are validated. We identify and resolve ambiguity in definitions for "sandbox", systematize ten years of sandbox research, and point out gaps in our current practices and propose ways forward in resolving them.

We contribute the following:

- A multi-disciplinary methodology for systematically analyzing the state of practice in a research domain ('Methodology').
- The first concise definition for "sandbox" that consistently describes research sandboxes ('What is a Sandbox').
- Systemization of the research sandboxing landscape ('Results').
- Identification of and proposed solutions to (1) an over-reliance on *ad hoc* arguments for security validation and (2) the neglect of sandbox and policy usability ('Strengthening Sandboxing Results').

## WHAT IS A SANDBOX?

In order to systematically analyze the "sandboxing" landscape we need to clarify the meaning of the term. We reviewed definitions used by practitioners and in papers within the field, both in the substance of the definitions and in their quality as definitions. This section reviews those definitions and establishes a definition for our use here, which we advance as an improved definition for the field.

A definition should be a concise statement of the exact meaning of a word and may be accompanied by narration of some properties implied by the definition. In this case, it

should clearly distinguish between mechanisms that are and are not sandboxes. To gain widespread use, a new definition must include all mechanisms that are already widely considered to be sandboxes.

In software security contexts, the term "sandboxing" has grown ambiguous. In an early published use, it described an approach for achieving fault isolation (*Wahbe et al., 1993*). Discussions where practicing programmers are trying to understand what sandboxing is often fail to achieve a precise resolution and instead describe the term by listing products that are typically considered to be sandboxes or cases where sandboxes are often used (http://stackoverflow.com/questions/2126174/what-is-sandboxing, http://security. stackexchange.com/questions/16291/are-sandboxes-overrated, http://en.wikipedia.org/ w/index.php?title=Sandbox_(computer_security)&oldid=596038515). However, we did find cases where attempts were made at a concise and general definition. A representative and accepted StackOverflow answer (http://security.stackexchange.com/questions/5334/ what-is-sandboxing) started with, "In the context of IT security, 'sandboxing' means isolating some piece of software in such a way that whatever it does, it will not spread havoc elsewhere"—a definition that is not sufficiently precise to separate sandboxes from other defensive measures.

Even recently published surveys of sandbox literature have either acknowledged the ambiguity, then used overly-broad definitions that include mechanisms not traditionally considered to be sandboxes (*Schreuders, McGill & Payne, 2013*), or have relied entirely on the use of examples instead of a precise definition (*Al Ameiri & Salah, 2011*). Schreuders writes, "Although the terminology in use varies, in general a sandbox is separate from the access controls applied to all running programs. Typically sandboxes only apply to programs explicitly launched into or from within a sandbox. In most cases no security context changes take place when a new process is started, and all programs in a particular sandbox run with the same set of rights. Sandboxes can either be permanent where resource changes persist after the programs finish running, or ephemeral where changes are discarded after the sandbox is no longer in use. ... " This definition suffers from three problems. First, it is still overly reliant on examples and thus is unlikely to capture all security mechanisms that are uncontroversially called sandboxes. Along the same lines, characterizations prefaced with, "In most cases ...", are not precise enough to reliably separate sandboxes from non-sandboxes. Finally, the comparison to access controls is not conclusive because it does not clarify which, if any, access control mechanisms applied to a subset of running programs are not sandboxes.

In this section we aim to resolve this ambiguity to lay the groundwork for our analysis's inclusion criteria. While this definition serves our purposes, we believe it can strengthen future attempts to communicate scientifically about sandboxes by adding additional precision. We derive a clear, concise definition for what a "sandbox" is using papers that appear in five top-tier security and operating system conferences, selected because their topics of interest are broad enough to include sandboxing papers most years. While we do not attempt to thoroughly validate our definition using commercial and open source sandboxes, it does encompass the tools with which we are most familiar.

We found 101 potential sandboxing papers. Out of these papers, 49 use the term "sandbox" at least once, and 14 provide either an explicit or implicit definition of the term that is clear enough to characterize. The remaining papers that use the term make no attempt at a definition or provide an ambiguous explanation, intertwined with other ideas, and spread over multiple sentences. Within the set of definitions we identify two themes: *sandboxing as encapsulation* and *sandboxing as policy enforcement.*

*Sandboxing as encapsulation* has a natural analogy: sandboxes on playgrounds provide a place for children to play with indisputably-defined bounds, making the children easier to watch, and where they are less likely to get hurt or hurt someone else. They also contain the sand, thus preventing it from getting strewn across neighboring surfaces. A similar analogy is used in an answer on the Security StackExchange to the question, "What is a sandbox?" Indeed, Wahbe was working to solve the problem of encapsulating software modules (to keep a fault in a distrusted module from affecting other modules) when he popularized the term in this domain.[1]

Table 1 lists the definitions we found that we characterize as falling within the theme of sandboxing as isolation. Many of these definitions use the term "isolation," but we prefer the use of *encapsulation*. In Object Oriented Programming, an object *encapsulates* related components and *selectively* restricts access to some of those components. Isolation, on the other hand, sometimes refers to a stronger property in which modules use entirely different resources and therefore cannot interfere with each other *at all*. Sandboxed components often need to cooperate to be useful. Cooperation and the idea of disjoint resources are present in Wahbe's original use of the term "sandbox": Wahbe was trying to reduce the communication overhead present in hardware fault isolation by instead creating software domains that run in the same hardware resources, but that do not interfere when faulty. One potential counterpoint to our use of "encapsulation" is that the term typically is used to refer to cases where the inside (e.g., of an object) is protected from the outside, but sandboxes often protect the external system from the contents of the sandbox. While this is a fair point, this paper does discuss sandboxes that protect their contents from the outside and sandboxes exist that simultaneously defend the inside from the outside and *vice versa* (Li et al., 2014). Furthermore, one can consider that a sandbox encapsulates an external system that must be protected from a potentially malicious component. Given these points, we maintain that encapsulation's recognition of cooperation is important enough to use the term over isolation. Nevertheless, we retain the use of *isolation* when discussing existing definitions.

Table 2 presents seven quotes that discuss sandboxing in terms of restrictions or policy enforcement. These definitions reflect different dimensions of the same idea: a *security policy* can state what is allowed, verboten, or both. The "sandbox" is the subject that enforces the policy or "sandboxing" is the act of enforcing a policy. In short, these quotes cast *sandboxing as policy enforcement.*

Careful inspection of our definition tables shows that the same technique, Sofware-based Fault Isolation (SFI), appears in both tables. Zhang explicitly states that hardening is not used in SFI, but McCamant very clearly refers to operations being "allowed" and the existence of a policy. While it could seem that the *sandboxing as isolation* and *sandboxing as*

[1] While it is clear from at least one publication that the term *sandbox* was used in computer security earlier than Wahbe's paper (*Neumann, 1990*), many early software protection papers cite Wahbe as the origin of the "sandbox" method (*Zhong, Edwards & Rees, 1997*; *Wallach et al., 1997*; *Schneider, 1997*). At least one early commentator felt that this use of the term "sandbox" was merely renaming "trusted computing bases" (TCB) (*McLean, 1997*). We believe this section makes it clear that sandboxes meet common TCB definitions, but that not all TCBs are sandboxes.

**Table 1** Definitions that speak about "sandboxing" in terms of isolation.

| Reference | Quote |
| --- | --- |
| *Zhang et al. (2013)* | "SFI (Software(-based) Fault Isolation) uses instruction rewriting but provides isolation (sandboxing) rather than hardening, typically allowing jumps anywhere within a sandboxed code region." |
| *Zeng, Tan & Erlingsson (2013)* | "It is a code-sandboxing technique that isolates untrusted modules from trusted environments. . . . In SFI, checks are inserted before memory-access and control-flow instructions to ensure memory access and control flow stay in a sandbox. A carefully designed interface is the only pathway through which sandboxed modules interact with the rest of the system." |
| *Geneiatakis et al. (2012)* | "Others works have also focused on shrinking the attack surface of applications by reducing the parts that are exposed to attack, and isolating the most vulnerable parts, using techniques like sandboxing and privilege separation." |
| *De Groef et al. (2012)* | "Isolation or sandboxing based approaches develop techniques where scripts can be included in web pages without giving them (full) access to the surrounding page and the browser API." |
| *Cappos et al. (2010)* | "Such sandboxes have gained widespread adoption with web browsers, within which they are used for untrusted code execution, to safely host plug-ins, and to control application behavior on closed platforms such as mobile phones. Despite the fact that program containment is their primary goal, flaws in these sandboxes represent a major risk to computer security." |
| *Reis et al. (2006)* | "Wagner et al. use system call interposition in Janus to confine untrusted applications to a secure sandbox environment." |
| *Cox et al. (2006)* | "Our work uses VMs to provide strong sandboxes for Web browser instances, but our contribution is much broader than the containment this provides." |

*policy enforcement* camps are disjoint, we claim they are talking about different dimensions of the same idea. Isolation refers to the *what*: an isolated environment where a module cannot do harm or be harmed. Policy enforcement refers to the *how*: by clearly defining what is or is not allowed. To use another childhood analogy, we often sandbox children when we place them in the corner as a punishment. We isolate them by moving them away from everyone else and placing them in a specific, bounded location, then we impose a security policy on them by making statements such as, "Do not speak, look straight ahead, and think about what you did." We resolve ambiguity in the use of the term "sandbox" by combining these themes:

**Sandbox**   An encapsulation mechanism that is used to impose a security policy on software components.

This definition concisely and consistently describes the research sandboxes we identify in the remainder of this paper.

**Table 2** Definitions that speak about "sandboxing" in terms of policy enforcement.

| Reference | Quote |
|---|---|
| *Xu, Saïdi & Anderson (2012)* | "We automatically repackage arbitrary applications to attach user-level sandboxing and policy enforcement code, which closely watches the applications behavior for security and privacy violations such as attempts to retrieve a users sensitive information, send SMS covertly to premium numbers, or access malicious IP addresses." |
| *Chandra et al. (2011)* | "The re-executed browser runs in a sandbox, and only has access to the client's HTTP cookie, ensuring that it gets no additional privileges despite running on the server." |
| *Politz et al. (2011)* | "ADsafe, like all Web sandboxes, consists of two inter-dependent components: (1) a static verifier, called JSLint, which filters out widgets not in a safe subset of JavaScript, and (2) a runtime library, adsafe.js, which implements DOM wrappers and other runtime checks." |
| *Tang, Mai & King (2010)* | "Fundamentally, rule-based OS sandboxing is about restricting unused or overly permissive interfaces exposed by today's operating systems." |
| *Sun et al. (2008)* | "Sandboxing is a commonly deployed proactive defense against untrusted (and hence potentially malicious) software. It restricts the set of resources (such as files) that can be written by an untrusted process, and also limits communication with other processes on the system." |
| *McCamant & Morrisett (2006)* | "Executing untrusted code while preserving security requires that the code be prevented from modifying memory or executing instructions except as explicitly allowed. Software-based fault isolation (SFI) or "sandboxing" enforces such a policy by rewriting the untrusted code at the instruction level." |
| *Provos (2003)* | "For an application executing in the sandbox, the system call gateway requests a policy decision from Systrace for every system call." |

## METHODOLOGY

In this section, we discuss the steps we took in order to select and analyze sandboxing papers and the sandboxes they describe. Our methodology is primarily based on the book "Qualitative Content Analysis in Practice" (QCA) (*Schreier, 2012*). *Barnes (2013)* provides a succinct summary of the methodology in Section 5.3 of his dissertation. This methodology originates in the social sciences (*Berelson, 1952*; *Krippendorff, 2013*; *Denzin & Lincoln, 2011*) and is intended to repeatably interpret qualitative data to answer a set of research questions. Figure 1 summarizes the iterative process we used to define our questions, pick and interpret papers ('Picking papers' and 'Categorizing the dataset'), and develop our results ('Analyzing the dataset').

QCA goes well beyond a systematic literature review (*Budgen & Brereton, 2006*; *Kitchenham et al., 2009*). While both QCA and systematic reviews require the definition of research questions and repeatable processes for collecting source material, reviews stop short of detailed analysis. QCA carries on where reviews end. When performing QCA, researchers define coding frames to clearly and repeatably establish how the source material will be interpreted to answer the research questions. The frames contain

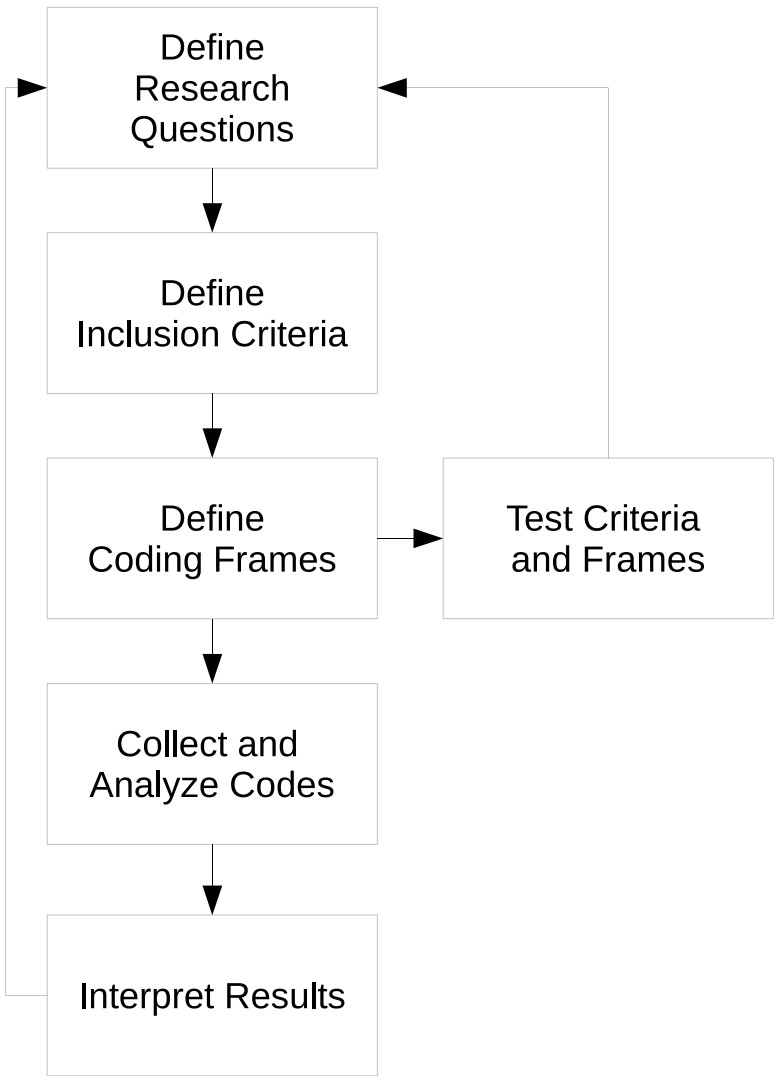

**Figure 1 The iterative process used to define research questions, build a dataset, and interpret the set to answer the questions.** This process is inspired by QCA (*Schreier, 2012*).

codes that summarize blocks of data and definitions for each code. Furthermore, QCA methodologies dictate how the coding frames are to be applied, by segmenting the entirety of the data such that each segment can labeled with at most one code. This ensures that the data is coded without missing relevant data and while reducing the researcher's bias towards some bits of data. Finally, QCA requires researchers to test their full process before carrying out the analysis.[2] Together, these steps allow researchers to reliably and effectively interpret text to answer research questions that are not possible to answer using a purely quantitative analysis. For example, Schreier points out that a quantitative analysis can determine how many women appear in magazine advertisements relative to men, but a qualitative analysis (e.g., QCA) is required to determine whether or not women are more likely to be placed within trivial contexts than men in those ads (*Schreier, 2012*, p. 2).

[2] We followed the QCA methodology specified by Schreier with one major deviation. We did not segment the text because the vast majority of the content in the papers is irrelevant to our needs. Most uses of QCA attempt to capture content of a text in its entirety. This was not our goal so we analyzed text more selectively.

The sandboxes we describe in this paper were selected from the proceedings of five conferences: IEEE Symposium on Security and Privacy (Oakland), Usenix Security, ACM Conference on Computer and Communications Security (CCS), ACM Symposium on Operating System Principles (SOSP), and Usenix Symposium on Operating System Design and Implementation (OSDI). We restricted our selection to particular conferences to improve reproducibility—because of this choice, the set of papers evaluated against our inclusion criteria is very well defined. To select these conferences, we collected all of the sandboxing papers we were aware of and the selected five venues contained far more sandboxing papers than any other venue.[3]

The selected conferences are widely regarded as the top-tier conferences in software security and operating systems (http://www.core.edu.au/index.php/conference-rankings, https://personal.cis.strath.ac.uk/changyu.dong/ranking.html, http://faculty.cs.tamu.edu/guofei/sec_conf_stat.htm, http://webdocs.cs.ualberta.ca/~zaiane/htmldocs/ConfRanking.html). Therefore, our data reflects the consensus of large communities.

Table 3 presents our twelve research questions, the areas each question attempts to illuminate, and a comprehensive list of their answers as manifested by our paper corpus. We derived an initial set of questions by considering which broad aspects of sandboxes are poorly understood and where better understanding may change how the community performs research in this space. As a result, the questions are necessarily biased by our own backgrounds and personal experiences. In particular, this led to an emphasis on questions about how mechanisms and policies are derived, applied, and evaluated. We added questions while we performed the analysis when we found that we had the data to answer new and interesting questions. Overall, these questions aim to capture a comprehensive snapshot of the current state of sandboxing research, with an emphasis on where sandboxes fit into the process of securing software systems, what policies are enforced and how they are defined and constructed, and what claims are made about sandboxes and how those claims are validated.

## Picking papers

We selected papers from 10 years worth of proceedings at the five conferences mentioned above. We decided whether a paper was included in our sample based on rigorous inclusion criteria so the process of including/excluding papers is repeatable. The most important criterion is that the paper describes a sandbox that meets the definition given in 'What is a Sandbox?'. The remaining criteria were added as we carried out the study to exclude papers that are incapable of answering the research questions and to clarify relevant nuances in the definition.

Papers were included if they met the following criteria:

- The paper documents the design of a novel tool or technique that falls under the *sandbox* definition
- The paper is a full conference paper
- The paper is about an instance of a sandbox (e.g., not a component for building new sandbox tools, theoretical constructs for sandboxes, etc.)

[3] Based on earlier criticism of this paper, we reevaluated our data set by looking at the past four years of proceedings at unselected venues such as the USENIX Annual Technical Conference (ATC), Programming Language Design and Implementation (PLDI), and Object-Oriented Programming, Systems, Languages and Applications (OOPSLA). These venues contained fewer sandboxing papers than our selected venues, and those that appeared were not significantly different in form or content from those in selected venues. In fact, with rare exceptions, the sandboxing papers at the unselected venues were written by the same authors as one or more paper in our data set.

**Table 3 Our research questions, the areas each question attempts to illuminate, and potential answers.** The answers are codes in the content analysis process we apply. Answers are not necessarily mutually exclusive. Definitions for the terms in this table appear in our coding frames (see Supplemental Information 1) with examples.

| Question area | Question | Possible answers |
|---|---|---|
| Sandbox lifecycle | Where in the architecture are policies enforced? | Component, application, host |
| | How and when are policies imposed? | Statically, dynamically, hybrid |
| Security outcomes | What resources do the sandboxes protect? | Memory, code/instructions, files, user data, communications |
| | Which components do the sandboxes protect? | Component, application, application class |
| | At what point will sandboxes catch exploits? | Pre-exploit, post-exploit |
| Effort and applicability | What must be done to apply the sandboxes? | Nothing, select pre-made policy, write policy, run tool, install tool |
| | What are the requirements on sandboxed components? | None, source code, annotated source code, special compiler, compiler-introduced metadata, sandbox framework/library components |
| Policy provenance and manifestation | Who defines policies? | Sandbox developer (fixed), sandbox user (user-defined), application developer (application-defined) |
| | How are policies managed? | Central policy repository, no management |
| | How are policies constructed? | Encoded in sandbox logic, encoded in application logic, user written |
| Research claims and validation | What claims are made about sandboxes? | Performance, security, applicability |
| | How are claims validated? | Proof, analytical analysis, benchmark suite, case studies, argumentation, using public data |
| | How are sandboxes released for review? | Source code, binaries, not available |

- Techniques are applied using some form of automation (e.g., not through entirely manual re-architecting)
- A policy is imposed on an identifiable category of applications or application subsets
  - The policy is imposed locally on an application (e.g., not on the principal the application executes as, not on network packets in-transit, etc.)
  - The category encompasses a reasonable number of real-world applications (e.g., doesn't require the use of (1) a research programming language, (2) extensive annotations, or (3) non-standard hardware)

We gathered papers by reading each title in the conference proceedings for a given year. We included a paper in our initial dataset if the title gave any indication that the paper could meet the criteria. We refined the criteria by reviewing papers in the initial dataset from Oakland before inspecting the proceedings from other venues. We read the remaining papers' abstracts, introductions, and conclusions and excluded papers as they were being interpreted if they did not meet the criteria. We maintained notes about why individual papers were excluded from the final set.[4]

## Categorizing the dataset

To interpret papers we developed coding frames[5] where a *category* is a research question and a *code* is a possible answer to the question. To ensure consistency in coding, our frames

[4] Our full list of papers with exclusion notes is available in Supplemental Information 2.

[5] Our full coding frames are available in Supplemental Information 1.

include detailed definitions and examples for each category and code. Our codes are not mutually exclusive: a question may have multiple answers. We developed the majority of our frames before performing a detailed analysis of the data, but with consideration for what we learned about sandboxing papers while testing the inclusion criteria above on our data from Oakland. We learned that evaluative questions were quite interesting while coding papers, thus frames concerning what claims were made about a sandbox and how those claims were validated became more fine-grained as the process progressed. Whenever we modified a frame, we updated the interpretations of all previously coded papers.

We tested the frames by having two coders interpret different subsets of the Oakland segment of the initial dataset. To interpret a paper, each category was assigned the appropriate code(s) and a quote justifying each code selection was highlighted and tagged in the paper's PDF.[6] While testing, the coders swapped quotes sans codes and independently re-assigned codes to ensure consistency, but we did not measure inter-rater reliability. Code definitions were revised where they were ambiguous. While there is still some risk that different coders would select different quotes or assign codes to the same quote, we believe our methodology sufficiently mitigated the risk without substantially burdening the process given the large scope of this effort.

After coding every paper, we organized the codes for each paper by category in a unified machine-readable file[7] (hereafter referred to as the summary of coded papers) for further processing.

## Analyzing the dataset

To summarize the differences and similarities between sandboxing papers, we attempted to identify clusters of similar sandboxing techniques. To do so, we first calculated a dissimilarity matrix for the sandboxes. For category $k$, let $p_{ijk}$ be the number of codes that sandboxes $i$ and $j$ share, divided by the total number of codes in that category they *could* share. For categories in which each sandbox is interpreted with one and only one code, $p_{ijk}$ is either 1 or 0; for other categories, it falls in the interval $[0, 1]$. Then the dissimilarity between $i$ and $j$ is $d_{ij} = \sum_k (1 - p_{ijk})$. We fed the resulting dissimilarity matrix into a hierarchical agglomerative clustering algorithm (*Kaufman & Rousseeuw, 2009*) (implemented in R with the `cluster` package (*R Core Team, 2014*; *Maechler et al., 2014*)). This algorithm begins by treating each sandbox as its own cluster, and then iteratively merges the clusters that are nearest to each other, where distance between two clusters is defined as the average dissimilarity between the clusters' members. The agglomerative clustering process is displayed in dendrograms. We stopped the agglomerative process at the point at which there were two clusters remaining, producing two lists of sandboxes, one list for each cluster. To interpret the resulting clusters, we produced bar charts displaying the code membership by cluster. We conducted this analysis three times: once using all of the categories to define dissimilarity, once using all categories except those for claims, validation, and availability, and once using the validation categories. We do not present the plots from the analysis that ignored claims, validation, and availability because it did not produce results different from those generated using all categories.

[6] A full list of quotes with code assignments is available in Supplemental Information 3.

[7] The summarized version of our dataset is available as Supplemental Information 4. This spreadsheet was converted to a CSV file to perform statistical and graph-based analyses.

We conducted correlational analyses to learn whether sandbox validation techniques have improved or worsened over time, or whether sandbox publications with better (or worse) validation received more citations. The validation codes were ordered in the following way: proof > analytical analysis > benchmarks > case study > argumentation > none. This ordering favors validation techniques that are less subjective. While it is possible for a highly ranked technique to be applied less effectively than a lower ranked technique (e.g., a proof that relies on unrealistic assumptions relative to a thorough case study) this ranking was devised after coding the papers and is motivated by the real world applications of each technique in our dataset. Each claim type (security, performance, and applicability), then, was an ordinal random variable, so rank-based methods were appropriate. When a sandbox paper belonged to two codes in a particular validation category, we used its highest-ordered code to define its rank, and lower-ordered codes to break ties. So, for instance, if paper A and paper B both included proofs, and paper A also included benchmarks, paper A would be ranked higher than paper B. To test if a claim type was improving over time, we estimated the Spearman correlation (*Spearman, 1904*) between its codes and the year of publication, and hence tested for a monotonic trend. Testing if papers with better validation, in a particular category, received more citations necessitated accounting for year of publication, since earlier papers typically have higher citation counts. To do so, we regressed paper citation rank against both publication year and category rank. (We used the rank of papers' citation counts as the dependent variable, as opposed to the citation counts themselves, due to the presence of an influential outlier—Terra (*Garfinkel et al., 2003*). Scatterplots show the relationship between citation ranks and publication year to be approximately linear, so a linear adjustment should suffice.) There was a "validation effect" if the coefficient on the validation measure was significantly different from zero. We conducted four separate regression analyses: one in which citation ranks were regressed on publication year and category ranks of all three validation criteria, and one in which citation ranks were regressed on publication year and security validation only, one in which citation ranks were regressed on publication year and performance validation only, and one in which citation ranks were regressed on publication year and applicability validation only.

We constructed a citation graph using the papers in our set as nodes and citations as edges as a final means of better understanding the sandboxing landscape. We clustered the nodes in this graph using the same clusters found statistically, using the process describe above, and using common topics of interest we observed. The topics of interest are typically based on the techniques the sandboxes apply (e.g., Control Flow Integrity (CFI), artificial diversity, etc.). We evaluate these clusters using the modularity metric, which enables us to compare the quality of the different categorizations. Modularity is the fraction of edges that lie within a partition, above the number that would be expected if edges were distributed randomly.

## RESULTS

We derived our results from the various statistical clusters of our summary of coded papers, trends explicit in this dataset, and observations made while reading the papers

or analyzing our summarized data. As our dataset is public, we encourage readers to explore the data themselves. Note while interpreting the statistical clusters that they are not representative of how papers are related in terms of broad topics of interest. When we applied the statistical clusters to the citation graph of the papers in our set the modularity scores were −0.04 and 0.02 when papers were clustered based on all of the attributes we coded and just validation attributes respectively. These modularity scores mean that the statistical clusters are no better than randomly clustering papers when considering how they cite each other.

These poor modularity scores make sense because authors are much more likely to cite papers that use similar techniques or tackle similar problems than use similar validation strategies. We confirmed the latter observation by computing that the modularity for overlapping groups (*Lázár, Ábel & Vicsek, 2009*) based on validation is −0.198, which confirms that partitions built from the validation techniques do not direct citation graph structure. Indeed, when we clustered papers in the citation graph based on topics of interest we observed while interpreting the set, the modularity score, 0.33, is significantly better than a random cluster. The citation graph with topic clusters is shown in Fig. 2. While these clusters are potentially of sociotechnical interest to the community, we must look at lower-level attributes to understand how sandboxes are to be applied in practice and how they improve the security posture of real systems. The statistical clusters fill that role.

Figures 3 and 4 show the codes that are members of the fixed policy and user-defined policy clusters respectively when all categories are considered. The dendrogram for these clusters appears in Fig. 5. Many of our results are interpretations of these charts. Table 4 succinctly describes our results per research question and references later sections where more details are found. The remainder of this section presents those details.

## Sandboxes: building materials for secure systems

Sandboxes are flexible security layers ready to improve the security posture of nearly any type of application. While the deployment requirements and details vary from sandbox to sandbox, collectively they can be applied at many different points in a system's architecture and may be introduced at any phase in an application's development lifecycle, starting with the initial implementation. In fact, sandboxes can even be applied well after an application has been abandoned by its maintainer to secure legacy systems.

In our dataset, the policy enforcement mechanism for a sandbox is always deployed as a system component, as a component of an application host, or by insertion directly into the component that is being encapsulated. While application hosts are becoming more popular as many applications are moved into web browsers and mobile environments, they are currently the least popular place to deploy policy enforcement mechanisms for research sandboxes. Our set includes ten sandboxes where policies are enforced in the application host, twenty-six in the component being encapsulated,[8] and thirty-two in a system component.

We believe that application hosts are less represented because many existing hosts come with a sandbox (e.g., the Java sandbox, Android's application sandbox, NaCl in Google

[8] *Sehr et al. (2010)* is counted twice because the enforcement mechanism is spread across the application and its host.

**Table 4  Summary of our research questions and results.**

| Research question | Results | Section |
|---|---|---|
| Where in a system's architecture are policies enforced? | There is an emphasis on enforcing policies in the operating system or transforming applications to enforce a policy over using application hosts (e.g., language-hosting virtual machines, browsers, etc.). | 'Sandboxes: building materials for secure systems' |
| When are policies imposed? | Static, dynamic, and hybrid strategies are roughly equally favored in all domains but with a slight preference for strictly static or dynamic approaches. | 'Sandboxes: building materials for secure systems' |
| What application resources are protected by sandboxes? | Sandboxes with fixed policies tend to prevent memory corruption or protect properties of application code (e.g., control flow). User-defined policies are correlated with policies that are more diverse and cover the gamut of application-managed resources. | 'Sandboxes: building materials for secure systems' |
| What types of components are protected by sandboxes? | Sandboxes that use fixed policies tend to require the user to target specific components, while those with user-defined policies tend to allow for broader targeting. | 'Sandboxes: building materials for secure systems' |
| At what point in the process of an attack will an exploit violate sandbox policies? | Sandboxes are primarily pro-active by disrupting exploits before a payload can be executed. Where users must define a policy, sandboxes tend to be pro-active in attempting to stop exploits, but also limit the range of possible behaviors a payload can exhibit. | 'Sandboxes: building materials for secure systems' |
| What are the requirements of people applying sandboxes? | Sandboxes that have fewer requirements for people tend to have more requirements for the application. Similarly, having a fixed policy is correlated with more requirements of the application, while user-defined policies are correlated with more requirements of the user. | 'Policy flexibility as a usability bellwether' |
| What are the requirements of components being sandboxed? | Sandboxes with fixed policies most-often require that applications be compiled using a special compiler. | 'Policy flexibility as a usability bellwether' |
| Who defines sandbox policies? | Policies are most often defined by the sandbox developer at design time. | 'Policy flexibility as a usability bellwether' |
| How are policies managed? | Policy management is largely ignored, even where users must write their own policies. | 'Policy flexibility as a usability bellwether' |
| How are policies constructed? | Most policies are hardcoded in the sandbox. | 'Policy flexibility as a usability bellwether' |
| What claims are made about sandboxes? | Applicability to new cases is often the impetus for improving existing techniques, but strong security and better performance are more often claimed. | 'The state of practice in sandbox validation' |
| How are claims validated? | Benchmarks and case studies are the most favored validation techniques for all types of claims. Where security claims are not validated using both benchmarks and case studies, ad-hoc arguments are heavily favored. | 'The state of practice in sandbox validation' |
| In what forms are sandboxes made available for review? | There is a recent slight increase in the release of sandbox source code, but generally no implementation artifacts are made available for review. | 'The state of practice in sandbox validation' |

Chrome, etc.). Indeed, all but one of the sandboxes deployed in application hosts are for the web, where applications can gain substantial benefits from further encapsulation and there is currently no *de facto* sandbox. The one exception is Robusta (*Siefers, Tan & Morrisett, 2010*), which enhances the Java sandbox to encapsulate additional non-web computations.

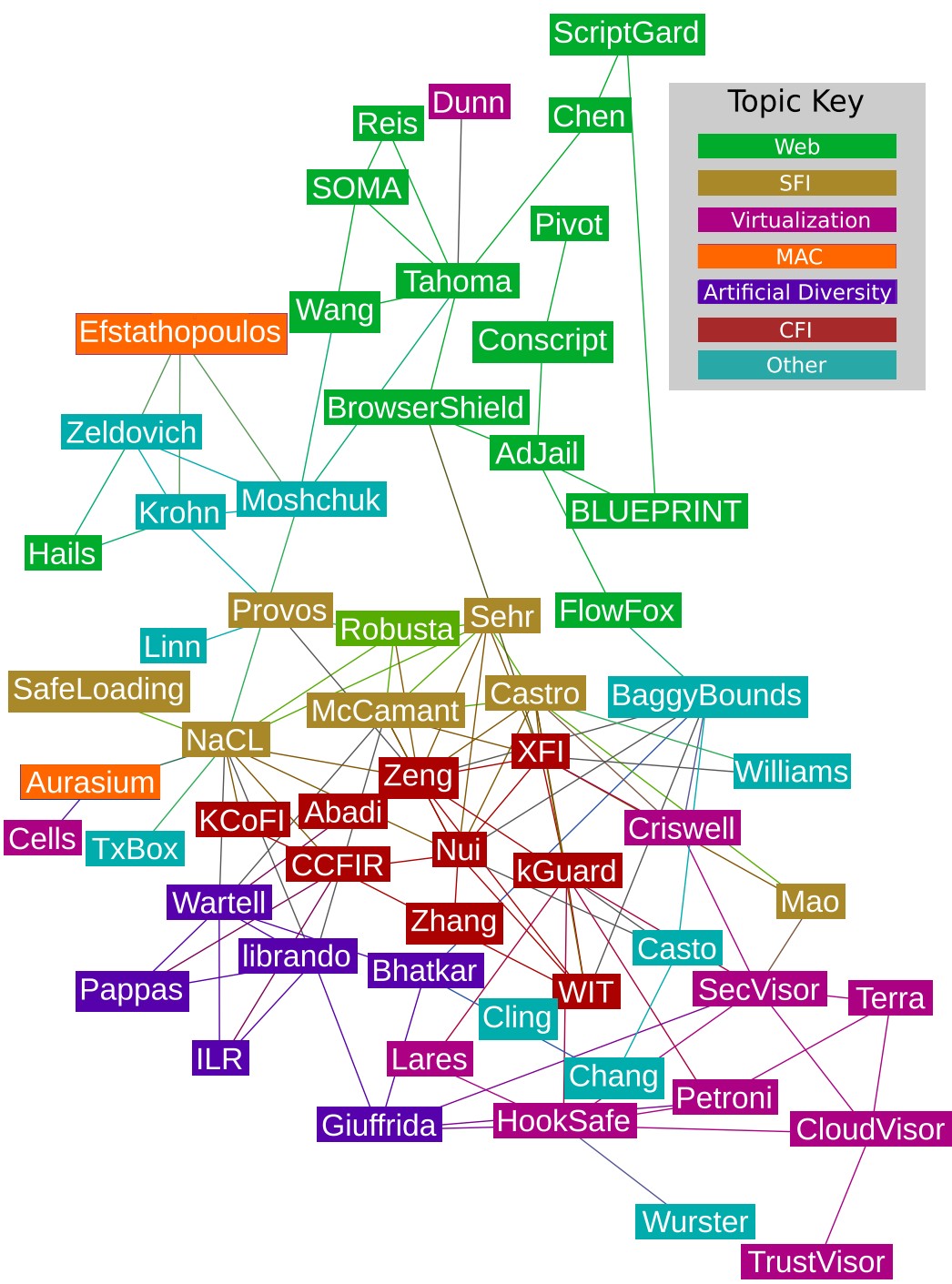

**Figure 2 The citation graph for the papers in our set.** The colors represent clusters based on topics of interest (modularity = 0.33). Papers cluster based on topics of interest, not necessarily their technical attributes or validation stratgies, thus we must look at lower level attributes to gain a broad understanding of the sandboxing landscape. Papers that were not linked to any of the other papers in the set are not shown. Categories bridging Mandatory Integrity and Access Control (MI/AC) were collapsed to simply Mandatory Access Control (MAC) for this graph. Our citation data can be found in Supplemental Information 5.

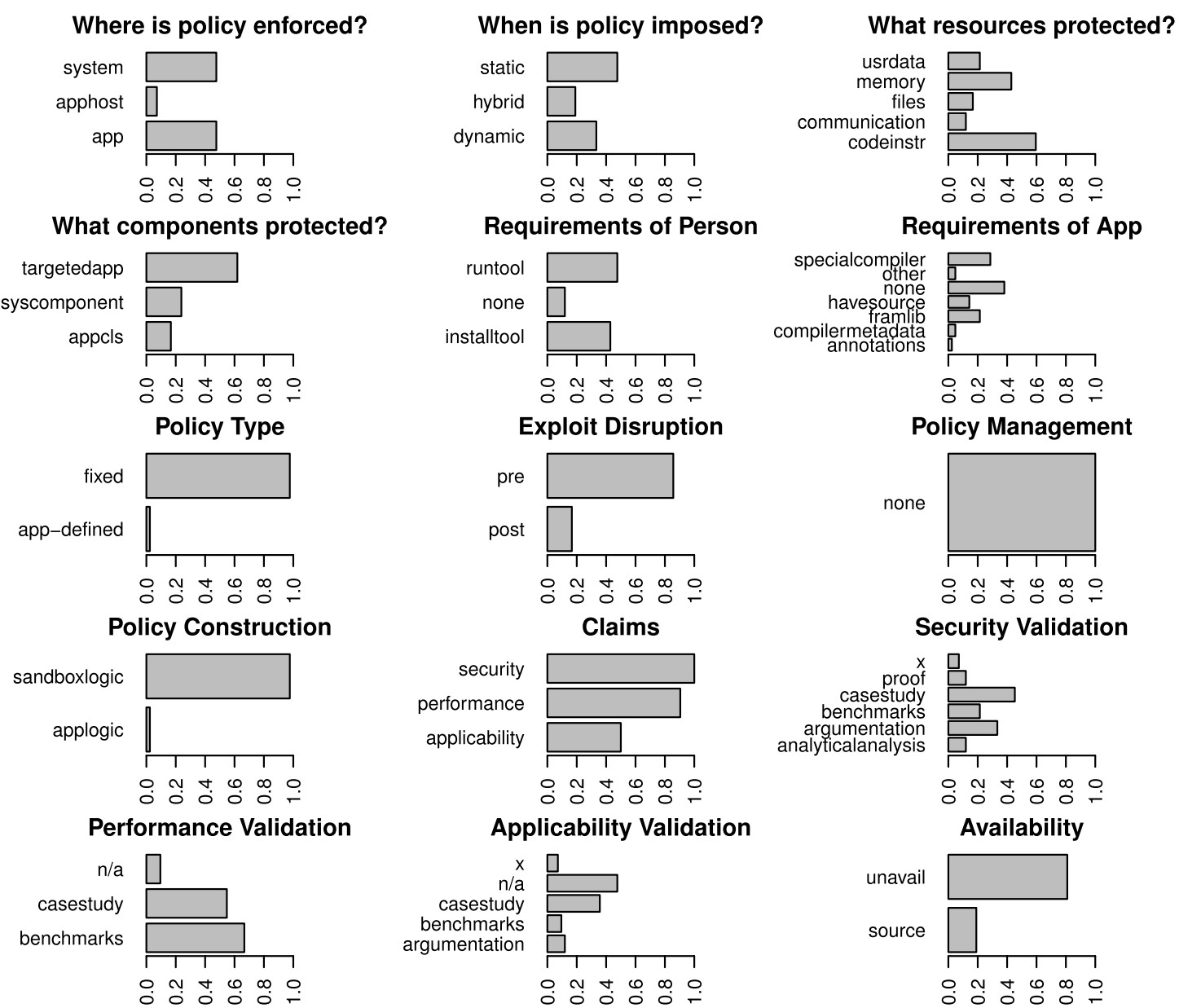

**Figure 3** Breakdown of the representation of all codes for papers that emphasize fixed policies. Cases where a claim was made but not validated are labeled with an "x".

System components are heavily represented because any sandbox that is to encapsulate a kernel, driver, or other system component must necessarily enforce the policy in a system component. Fifteen of the sandboxes fall into this category because they are encapsulating either a kernel or hypervisor. The remainder could potentially enforce their policies from a less privileged position, but take advantage of the full access to data and transparency to user-mode applications available to system components. This power is useful when enforcing information flow across applications, when preventing memory corruption, or when otherwise enforcing the same policy on every user-mode application.

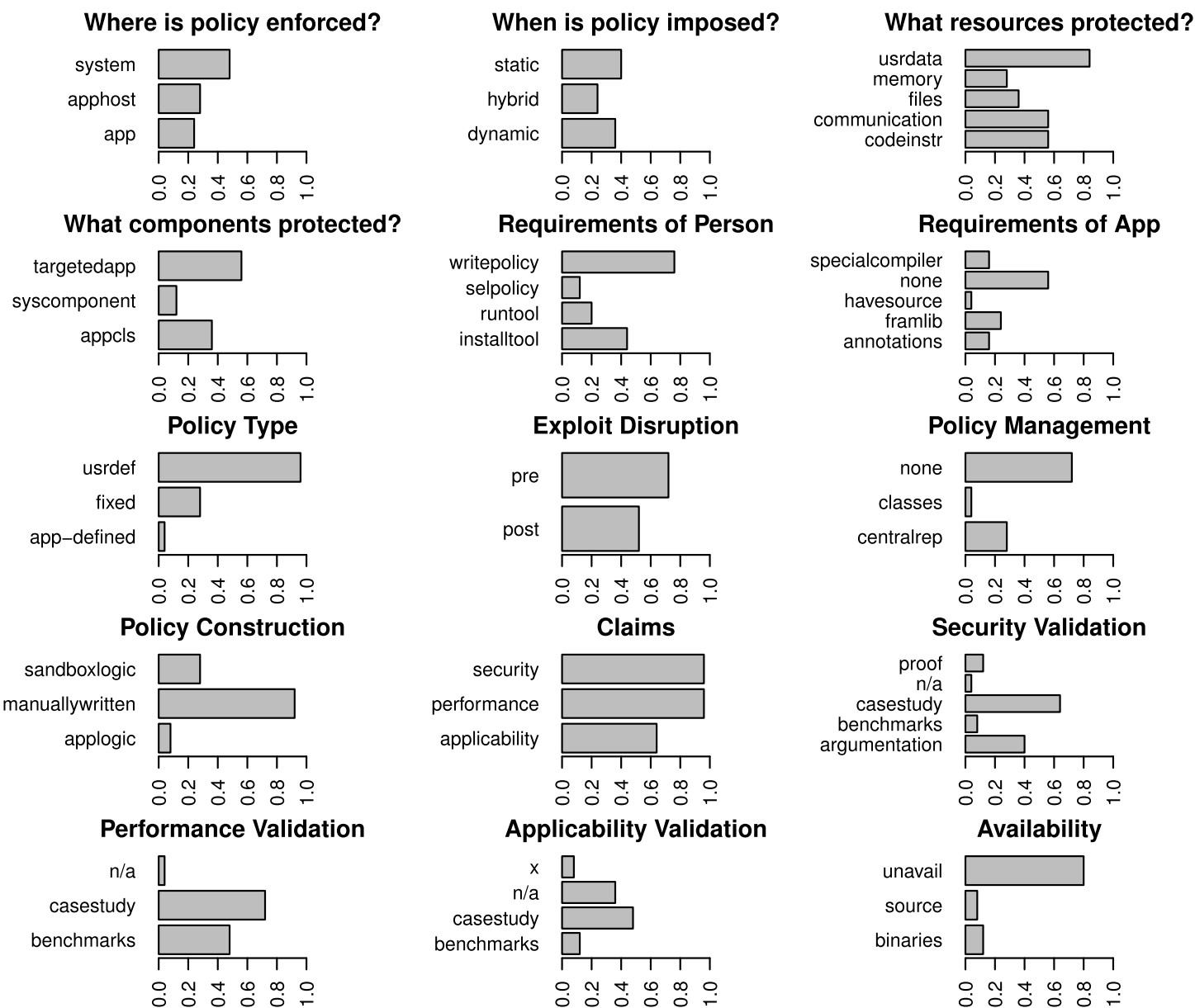

**Figure 4** **Breakdown of the representation of all codes for papers that emphasize user-defined policies.** Some sandboxes support a fixed-policy with an optional user-defined policy (e.g., *Siefers, Tan & Morrisett, 2010*). Cases where a claim was made but not validated are labeled with an "x".

Research sandboxes almost universally embed their enforcement mechanism in the application that is being encapsulated when the application runs in user-mode. Application deployment is correlated with fixed policies where modifying the application itself can lead to higher performance and where it makes sense to ensure the enforcement mechanisms exist anywhere the application is, even if the application moves to a different environment. Fixed-policies with embedded enforcement mechanisms are correlated with another important deployment concern: statically imposed policies.

Imposing a policy statically, most often using a special compiler or program re-writer, is advantageous because the policy and its enforcement mechanism can travel with the ap-

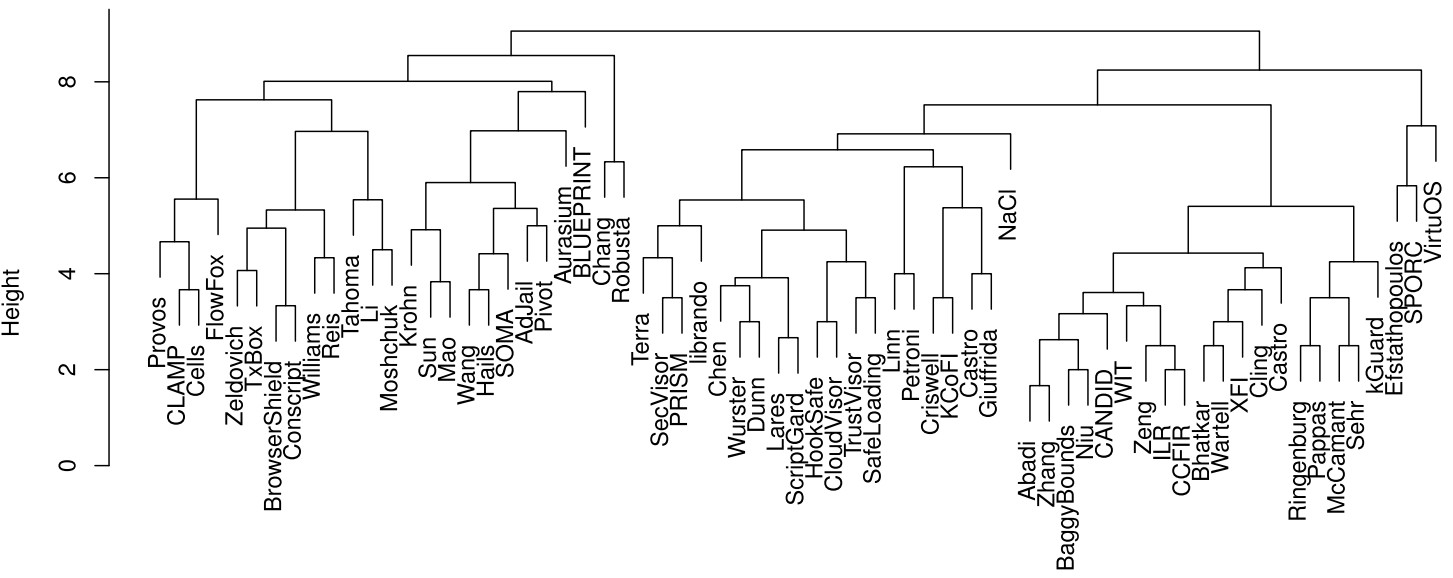

**Dendrogram of agnes(x = distanceMatrix, diss = TRUE)**

distanceMatrix
Agglomerative Coefficient = 0.57

**Figure 5  A dendrogram displaying the clusters for sandboxing papers taking into account all categories.** At the topmost level, where two clusters exist, the clusters respectively represent sandboxes that use fixed policies and those that use user-defined policies.

plication and overhead can be lower as enforcement is tailored to the targeted code. There are some cons to this approach. For example, the process of imposing the policy cannot be dependent on information that is only available at run-time and the policy is relatively unadaptable after it is set. Furthermore, because the policies are less adaptable, sandboxes that statically impose security policies typically only encapsulate components that are targeted by the person applying the sandbox. These are cases where dynamic mechanisms shine. Given these trade-offs, it makes sense that papers in our set fall into one of two clusters when all codes are considered: those that are protecting memory and software code, which are relatively easy to encapsulate with a fixed policy, and those managing behaviors manifested in external application communications or interactions with user-data and files that are more easily encapsulated with an adaptable (typically user-defined) policy.

Generally hybrid deployments are used when the approach is necessarily dynamic but static pre-processing lowers overhead. Sometimes, techniques begin as hybrid approaches and evolve to fully dynamic approaches as they gain traction. For example, early papers that introduce diversity in binaries to make reliable exploits harder to write (e.g., code randomization), tend to rely on compiler-introduced metadata, while later papers did not need the extra help. This evolution broadens the applicability of the sandboxing technique. We observed other techniques such as SFI and CFI evolve by reducing the number of requirements on the application, the person applying the sandbox, or both.

## Policy flexibility as a usability bellwether

Requiring more work out of the user or more specific attributes of an application lowers the odds that a sandbox will be applied, thus it is natural that research on specific techniques reduce these burdens over time. We find that the nature of the policy has an influence on how burdensome a sandbox is. About half of sandboxes with fixed policies require the application be compiled using a special compiler or uses a sandbox-specific framework or library. Many fixed-policy sandboxes also require the user to run a tool, often a program re-writer, or to install some sandbox component. In comparison, nearly all sandboxes with flexible policies require the user to write a policy manually, but few have additional requirements for the application. Given the burdens involved in manually writing a security policy, the message is clear—easy to use sandboxes reduce the user-facing flexibility of the policies they impose.

Forty-eight sandboxes, more than two-thirds of our sample, use a fixed policy. In all of these cases the policy itself exists within the logic of the sandbox. In the remaining cases, the policy is encoded in the logic of the application twice (e.g., through the use of the sandbox as a framework), and the remaining seventeen cases require the user to manually write a policy.

In cases where the user must manually write the policy, it would help the user if the sandbox supported a mechanism for managing policies—to ensure policies do not have to be duplicated repeatedly for the same application, to generate starter policies for specific cases, to ensure policies can apply to multiple applications, etc. This type of management reduces the burden of having to manually write policies in potentially complex custom policy languages. Support for the policy writer is also important because the policies themselves can be a source of vulnerabilities (*Rosenberg, 2012*). Eight out of twenty-six cases where policy management is appropriate offered some central mechanism for storing existing policies, where they could potentially be shared among users. However, none of the papers in our sample list policy management as a contribution, nor do any of the papers attempt to validate any management constructs that are present. However, it is possible that there are papers outside of our target conferences that explicitly discuss management. For example, programming languages and software engineering conferences are more focused on policy authoring concerns and management may therefore be the focus of a paper that appears in one of those conferences. However, in spite of the fact that two of the authors of this paper are active researchers in the Programming Language community and three are active in the Software Engineering community, we are not aware of any such paper.

## The state of practice in sandbox validation

There is little variation in the claims that are made about sandboxes. Most claim to either encapsulate a set of threats or to increase the difficulty of writing successful exploits for code-level vulnerabilities. All but four measure the performance overhead introduced by the sandbox. Thirty-seven papers, more than half, make claims about the types of components the sandbox applies to, typically because the paper applies an existing technique to a different domain or extends it to additional components.

While there is wide variety in how these claims are validated, we observe measurable patterns. In our data set, proof and analytical analysis were, by far, the least used techniques. The lack of analytical analysis is due to the fact that the technique is primarily useful when the security of the mechanism depends on randomness, which is true of few sandboxes in our set. However, proof appears in two cases: (1) to prove properties of data flows and (2) six papers prove the correctness of a mechanism enforcing a fixed policy. The rarity of proof in the sandboxing domain is not surprising given the difficulty involved. Proof is particularly difficult in cases where one would ideally prove that a policy enforcement mechanism is capable of enforcing all possible policies a user can define, which we did not see attempted. Instead, claims are often validated empirically or in ways that are *ad hoc* and qualitative.

In empirical evaluations, case studies are the most common technique for all claims, often because proof was not attempted and there is no existing benchmark suite that highlights the novel aspects of the sandbox. For example, papers for sandboxes with fixed policies often want to show a particular class of vulnerabilities can no longer be exploited in sandboxed code, thus examples of vulnerable applications and exploits for their vulnerabilities must be gathered or, very rarely, synthesized. When claims were empirically validated, the results were not comparable in fifteen out of sixty-two cases for performance, twenty-two out of forty-two cases for security, and twenty-four out of thirty-one cases for applicability because non-public data was used in the discussed experiments. Non-public data takes the form of unlabeled exploits, undisclosed changes to public applications, and unreleased custom example cases (e.g., applications built using a sandbox's framework where the examples were not released).

Security claims are notoriously difficult to formalize, hence the pervasive lack of proof. Many papers instead vet their security claims using multi-faceted strategies, often including both common empirical approaches: case studies and experiments using benchmark suites. However, Figs. 6 and 7 illustrate an interesting finding: in twenty-nine papers where multi-faceted strategies are not used, authors pick one empirical tactic and argue that their claims are true. Argumentation in this space is problematic because all of the arguments are *ad hoc*, which makes evaluations that should be comparable difficult to compare at best but more often incomparable. Furthermore, we observed many cases where arguments essentially summarize as, "Our sandbox is secure because the design is secure," with details of the design occupying most of the paper in entirely qualitative form. Not only are these types of arguments difficult to compare in cases where sandboxes are otherwise quite similar, it is even harder to see if they are complete in the sense that every sub-claim is adequately addressed.

Our correlational analyses show no significant trends in security or applicability analyses, however performance validation has improved over time. Table 5 summarizes the Spearman correlations and their $p$-values per validation category. Spearman correlations fall in the range $[-1, 1]$, where a value of 0 is interpreted as no correlation, positive values show a positive correlation, and negative values a negative correlation. The magnitude of the coefficient grows towards 1 as time and the validation rank become closer to perfect

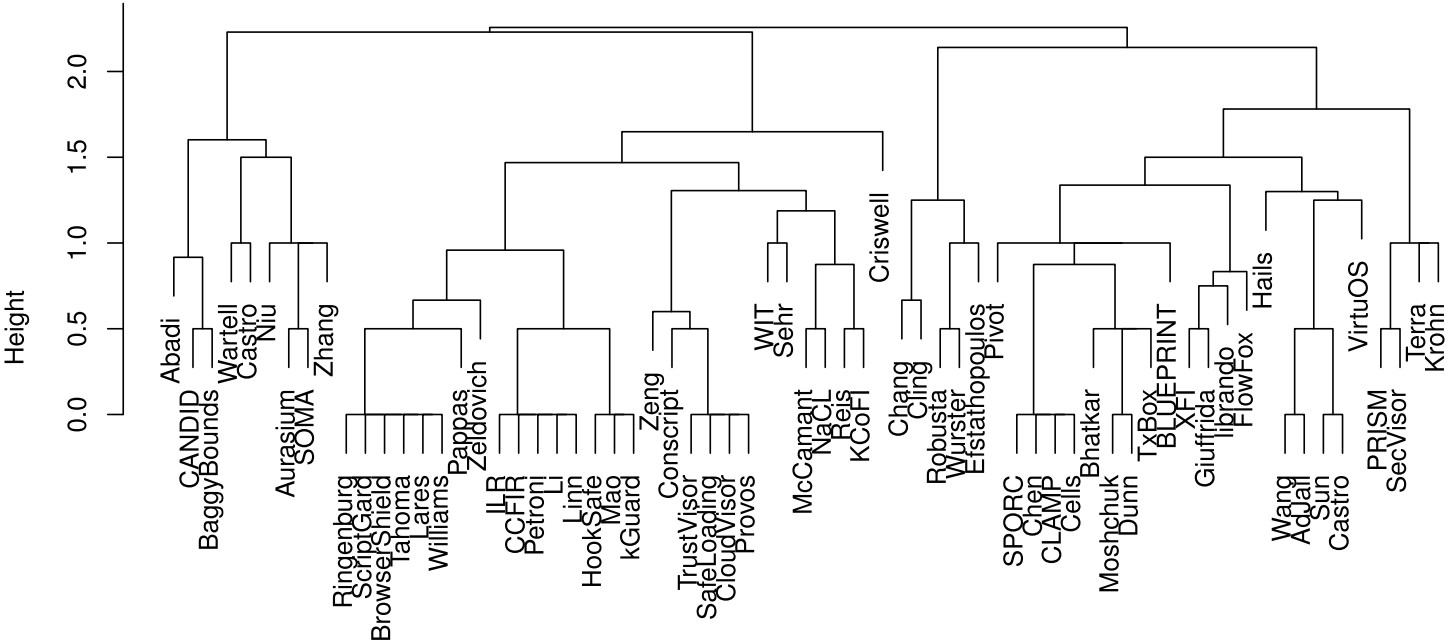

**Figure 6 A dendrogram displaying the clusters for sandboxing papers taking into account validation categories.** At the topmost level, where two clusters exist, the clusters respectively represent sandboxes that emphasize multi-faceted empirical security validation and those that do not.

monotonic functions (i.e., when a positive and perfect monotonic relationship exists, the Spearman correlation is 1).

Performance validation is positively, and statistically significantly, correlated with the passage of time. We observe that performance validation has advanced from a heavy reliance on benchmark suites to the use multi-faceted strategies that include the use of benchmark suites and case studies (typically to perform micro-benchmarks) that make use of public data—which ensures the results are comparable with future sandboxes. While the applicability validation correlation is not statistically significant, we observe that argumentation was abandoned early on in favor of case studies, with some emphasis on including benchmark suites in later years. There is no apparent change in security validation over time.

We fit linear models to each validation category separately and together relative to ranked citation counts to see if validation practices are predictive of future citations. All of the models achieved an $R$-squared value of 0.54 which suggests that passage of time and validation practices jointly explain about half of the variance in citation count ranks. Validation practices on their own are not predictive of how highly cited a paper will

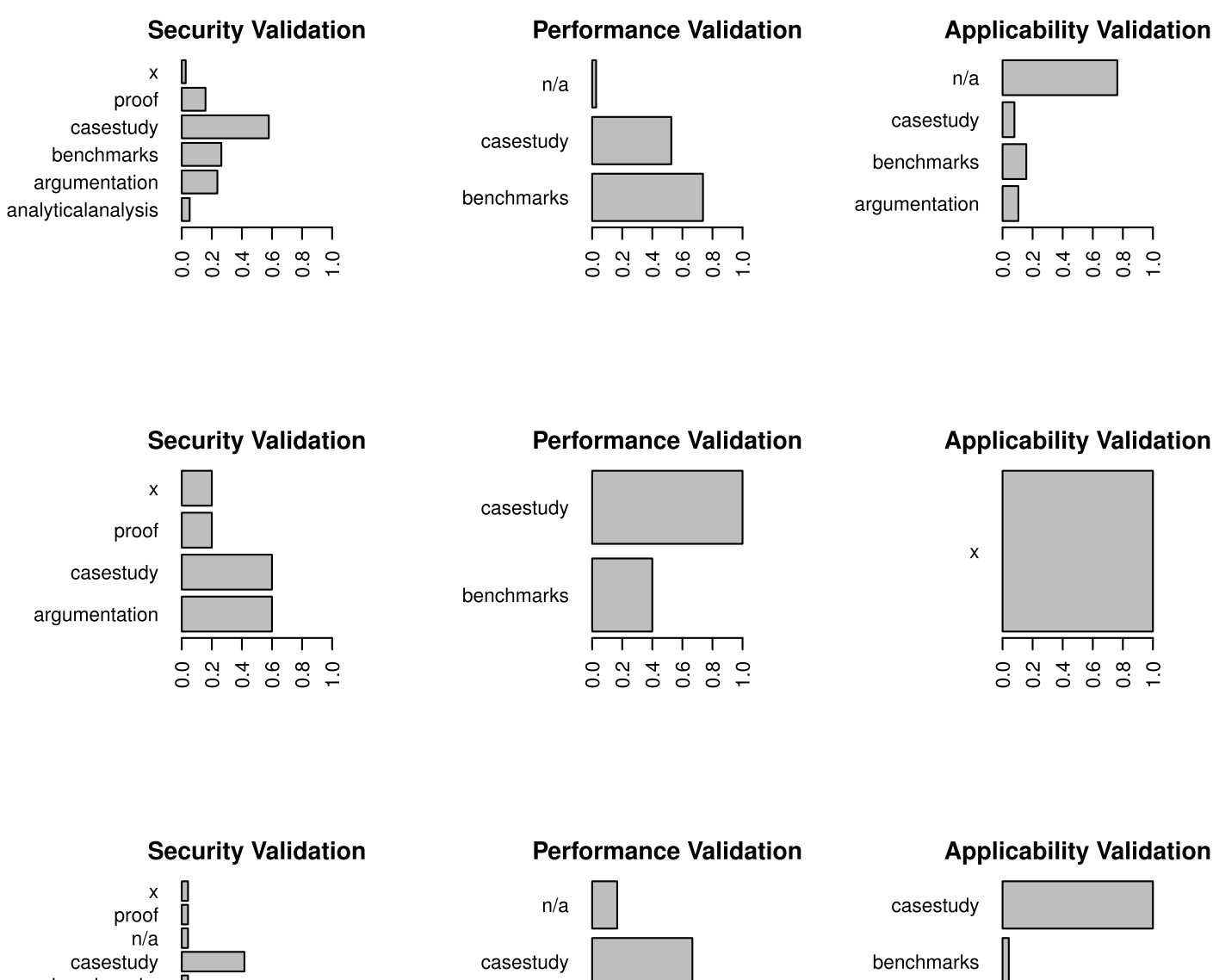

**Figure 7** **Breakdown of the representation of validation codes per claim type for the three validation clusters found in our dataset.** Each row contains the data for one cluster. The bottom two clusters include papers that do not emphasize multi-faceted security validation strategies, instead relying on case studies and arguments that security claims are true. Cases where a claim was made but not validated are labeled with an "x".

become. Table 6 summarizes the types of claims and the validation strategies employed per type for each paper in our set.

## STRENGTHENING SANDBOXING RESULTS

The existing body of knowledge within the sandboxing community provides a strong basis for securing current and future software systems. However, the results in 'Results' highlight

**Table 5 The Spearman correlations and their statistical significances per validation category.** Data with correlation coefficients closer to 1 have stronger correlations.

|  | Correlation ($\rho$) | $p$-value |
|---|---|---|
| Security validation | −0.02 | 0.894 |
| Performance validation | 0.30 | 0.014 |
| Applicability validation | 0.20 | 0.105 |

several gaps. In this section we discuss how structured arguments can solve the problems presented by incomparable and incomplete *ad hoc* arguments ('Structured arguments') and possible ways to enhance sandbox and policy usability (Sandbox and policy usability).

## Structured arguments

Sandboxes are often evaluated against coarse criteria such as the ability to stop exploits against certain classes of vulnerabilities, to encapsulate certain categories of operations, or to function in new environments. However, these coarse criteria typically require the sandbox to address a number of sub-criteria. For example, *Zhang & Sekar (2013)* provide CFI without requiring compiler support or *a priori* metadata, unlike earlier implementations. To ensure the technique is secure, they must be sure that independently transformed program modules maintain CFI when composed. Details that clarify how an individual criterion is fulfilled can easily be lost when *ad hoc* arguments are used in an effort to persuade readers that the criterion has been met; particularly in sandboxes with non-trivial design and implementation details. This can leave the reader unable to compare similar sandboxes or confused about whether or not contributions were validated.

Since many of the security criteria are repeated across most papers, the cost of developing substructure can be amortized across lots of communal use. There are many possible ways to structure arguments in support to security claims:

- Assurance cases (*Weinstock, Lipson & Goodenough, 2007*; *Kelly, 1999*) provide graphical structures that explicitly tie claims together in trees that show how claims are narrowed. *Knight (2015)* provides a concise introduction to the topic. These structures also explicitly link leaf claims to the evidence that supports the claim. Assurance cases were created in response to several fatal accidents resulting from failures to systematically and thoroughly understand safety concerns in physical systems. Their use has spread to security and safety critical systems of nearly every variety in recent decades with case studies from aerospace (*Graydon, Knight & Strunk, 2007*) and a sandbox called S³ (*Rodes et al., 2015*) that was not analyzed as part of this study (*Nguyen-Tuong et al., 2014*). Sandboxing papers can use assurance cases to decompose claims to their most simple components, then link those components to relevant evidence in the paper (e.g., a summary of specific results, a specific section reference, etc.).
- *Maass, Scherlis & Aldrich (2014)* use a qualitative framework to compare sandboxes based on what happens when a sandbox fails, is bypassed, or holds. Authors could

Table 6 Claims made about sandboxes (●: Security, ✍: Performance, and ✂: Applicability) and their validation strategies (♞: Proof, ✂: Analytical Analysis, ⟳: Benchmarks, ⌂: Case Studies, and ⚇: Argumentation). Grayed out icons mean a claim was not made or a strategy was not used. Icons made by Freepik from www.flaticon.com.

| Category | Citation | Conference | Claims | ● Val. | ✍ Val. | ✂ Val. |
|---|---|---|---|---|---|---|
| Other (Syscall) | *Provos (2003)* | Usenix | ●✍✂ | ♞✂⟳⌂⚇ | ⟳⌂ | |
| Virtualization | *Garfinkel et al. (2003)* | SOSP | ●✍✂ | ♞✂⟳⌂⚇ | | ⟳⌂⚇ |
| Diversity | *Bhatkar, Sekar & DuVarney (2005)* | Usenix | ●✍✂ | ♞✂⟳⌂⚇ | ⟳⌂ | ⟳⌂⚇ |
| Other (Syscall) | *Linn et al. (2005)* | Usenix | ●✍✂ | ♞✂⟳⌂⚇ | ⟳⌂ | |
| CFI | *Abadi et al. (2005)* | CCS | ●✍✂ | ♞✂⟳⌂⚇ | ⟳⌂ | ⟳⌂⚇ |
| Other (Memory) | *Ringenburg & Grossman (2005)* | CCS | ●✍✂ | ♞✂⟳⌂⚇ | ⟳⌂ | |
| MAC | *Efstathopoulos et al. (2005)* | SOSP | ●✍✂ | ♞✂⟳⌂⚇ | ⟳⌂ | |
| Web | *Cox et al. (2006)* | Oakland | ●✍✂ | ♞✂⟳⌂⚇ | ⟳⌂ | |
| SFI | *McCamant & Morrisett (2006)* | Usenix | ●✍✂ | ♞✂⟳⌂⚇ | ⟳⌂ | |
| CFI, SFI | *Erlingsson et al. (2006)* | OSDI | ●✍✂ | ♞✂⟳⌂⚇ | ⟳⌂ | ⟳⌂⚇ |
| Other (DFI) | *Castro, Costa & Harris (2006)* | OSDI | ●✍✂ | ♞✂⟳⌂⚇ | ⟳⌂ | ⟳⌂⚇ |
| Web | *Reis et al. (2006)* | OSDI | ●✍✂ | ♞✂⟳⌂⚇ | ⟳⌂ | |
| Other (InfoFlow) | *Zeldovich et al. (2006)* | OSDI | ●✍✂ | ♞✂⟳⌂⚇ | ⟳⌂ | |
| MI/AC | *Li, Mao & Chen (2007)* | Oakland | ●✍✂ | ♞✂⟳⌂⚇ | ⟳⌂ | |
| Web | *Bandhakavi et al. (2007)* | CCS | ●✍✂ | ♞✂⟳⌂⚇ | ⟳⌂ | ⟳⚇ |
| Web | *Chen, Ross & Wang (2007)* | CCS | ●✍✂ | ♞✂⟳⌂⚇ | ⟳⌂ | ⟳⌂⚇ |
| Virtualization | *Petroni & Hicks (2007)* | CCS | ●✍✂ | ♞✂⟳⌂⚇ | ⟳⌂ | |
| Virtualization | *Seshadri et al. (2007)* | SOSP | ●✍✂ | ♞✂⟳⌂⚇ | | ⟳⌂⚇ |
| Virtualization | *Criswell et al. (2007)* | SOSP | ●✍✂ | ♞✂⟳⌂⚇ | | |
| Web | *Wang et al. (2007)* | SOSP | ●✍✂ | ♞✂⟳⌂⚇ | ⟳⌂ | ⟳⌂⚇ |
| Other (InfoFlow) | *Krohn et al. (2007)* | SOSP | ●✍✂ | ♞✂⟳⌂⚇ | | ⟳⌂⚇ |
| CFI | *Akritidis et al. (2008)* | Oakland | ●✍✂ | ♞✂⟳⌂⚇ | ⟳⌂ | |
| Virtualization | *Payne et al. (2008)* | Oakland | ●✍✂ | ♞✂⟳⌂⚇ | ⟳ ⌂ | |
| MI/AC | *Sun et al. (2008)* | Oakland | ●✍✂ | ♞✂⟳⌂⚇ | ⟳⌂ | ⟳⌂⚇ |
| Other (TaintTrack) | *Chang, Streiff & Lin (2008)* | CCS | ●✍✂ | ♞✂⟳⌂⚇ | ⟳⌂ | ⟳⌂⚇ |
| Web | *Oda et al. (2008)* | CCS | ●✍✂ | ♞✂⟳⌂⚇ | ⟳⌂ | ⟳⌂⚇ |
| Other (OS) | *Williams et al. (2008)* | OSDI | ●✍✂ | ♞✂⟳⌂⚇ | ⟳⌂ | |
| SFI | *Yee et al. (2009)* | Oakland | ●✍✂ | ♞✂⟳⌂⚇ | ⟳⌂ | |
| Web | *Louw & Venkatakrishnan (2009)* | Oakland | ●✍✂ | ♞✂⟳⌂⚇ | ⟳⌂ | ⟳⌂⚇ |
| Web | *Parno et al. (2009)* | Oakland | ●✍✂ | ♞✂⟳⌂⚇ | ⟳⌂ | ⟳⌂⚇ |
| Other (Memory) | *Akritidis et al. (2009)* | Usenix | ●✍✂ | ♞✂⟳⌂⚇ | ⟳⌂ | ⟳⌂⚇ |
| Virtualization | *Wang et al. (2009)* | CCS | ●✍✂ | ♞✂⟳⌂⚇ | ⟳⌂ | |
| SFI | *Castro et al. (2009)* | SOSP | ●✍✂ | ♞✂⟳⌂⚇ | ⟳⌂ | ⟳⌂⚇ |
| Virtualization | *McCune et al. (2010)* | Oakland | ●✍✂ | ♞✂⟳⌂⚇ | ⟳⌂ | |
| Web | *Meyerovich & Livshits (2010)* | Oakland | ●✍✂ | ♞✂⟳⌂⚇ | ⟳⌂ | |
| Other (Memory) | *Akritidis (2010)* | Usenix | ●✍✂ | ♞✂⟳⌂⚇ | ⟳⌂ | ⟳⌂⚇ |
| SFI | *Sehr et al. (2010)* | Usenix | ●✍✂ | ♞✂⟳⌂⚇ | ⟳⌂ | |
| Web | *Louw, Ganesh & Venkatakrishnan (2010)* | Usenix | ●✍✂ | ♞✂⟳⌂⚇ | ⟳⌂ | ⟳⌂⚇ |
| Other (OS) | *Wurster & van Oorschot (2010)* | CCS | ●✍✂ | ♞✂⟳⌂⚇ | ⟳⌂ | ⟳⌂⚇ |
| SFI, Other (UserPolicy) | *Siefers, Tan & Morrisett (2010)* | CCS | ●✍✂ | ♞✂⟳⌂⚇ | ⟳⌂ | ⟳⌂⚇ |

Table 6 (*continued*)

| Category | Citation | Conference | Claims | ♥ Val. | 🏃 Val. | 📢 Val. |
|---|---|---|---|---|---|---|
| Web | *Feldman et al. (2010)* | OSDI | [icons] | [icons] | [icons] | [icons] |
| MI/AC | *Owen et al. (2011)* | Oakland | [icons] | [icons] | | [icons] |
| Other (Transactions) | *Jana, Porter & Shmatikov (2011)* | Oakland | [icons] | [icons] | [icons] | [icons] |
| CFI | *Zeng, Tan & Morrisett (2011)* | CCS | [icons] | [icons] | [icons] | |
| Web | *Saxena, Molnar & Livshits (2011)* | CCS | [icons] | [icons] | [icons] | |
| Web | *Chen et al. (2011)* | CCS | [icons] | [icons] | [icons] | |
| Virtualization | *Zhang et al. (2011)* | SOSP | [icons] | [icons] | [icons] | |
| SFI | *Mao et al. (2011)* | SOSP | [icons] | [icons] | [icons] | |
| Virtualization | *Andrus et al. (2011)* | SOSP | [icons] | [icons] | [icons] | [icons] |
| Diversity | *Pappas, Polychronakis & Keromytis (2012)* | Oakland | [icons] | [icons] | [icons] | |
| Diversity | *Hiser et al. (2012)* | Oakland | [icons] | [icons] | [icons] | |
| SFI | *Payer, Hartmann & Gross (2012)* | Oakland | [icons] | [icons] | [icons] | |
| CFI | *Kemerlis, Portokalidis & Keromytis (2012)* | Usenix | [icons] | [icons] | [icons] | |
| Diversity | *Giuffrida, Kuijsten & Tanenbaum (2012)* | Usenix | [icons] | [icons] | [icons] | [icons] |
| MI/AC | *Xu, Saïdi & Anderson (2012)* | Usenix | [icons] | [icons] | [icons] | [icons] |
| Diversity | *Wartell et al. (2012)* | CCS | [icons] | [icons] | [icons] | [icons] |
| Web, Other (InfoFlow) | *De Groef et al. (2012)* | CCS | [icons] | [icons] | [icons] | [icons] |
| Virtualization | *Dunn et al. (2012)* | OSDI | [icons] | [icons] | [icons] | [icons] |
| Web (MI/AC) | *Giffin et al. (2012)* | OSDI | [icons] | [icons] | [icons] | [icons] |
| CFI | *Zhang et al. (2013)* | Oakland | [icons] | [icons] | [icons] | |
| CFI | *Zhang & Sekar (2013)* | Usenix | [icons] | [icons] | [icons] | [icons] |
| CFI, SFI | *Niu & Tan (2013)* | CCS | [icons] | [icons] | [icons] | [icons] |
| Diversity | *Homescu et al. (2013)* | CCS | [icons] | [icons] | [icons] | [icons] |
| Other (OS) | *Moshchuk, Wang & Liu (2013)* | CCS | [icons] | [icons] | [icons] | [icons] |
| Virtualization | *Nikolaev & Back (2013)* | SOSP | [icons] | [icons] | [icons] | [icons] |
| CFI | *Criswell, Dautenhahn & Adve (2014)* | Oakland | [icons] | [icons] | [icons] | |
| Web | *Mickens (2014)* | Oakland | [icons] | [icons] | [icons] | [icons] |

structure their arguments by using the framework to describe their specific sandbox without performing explicit comparisons.

- Structured abstracts (*Hartley, 2004*; *Haynes et al., 1990*) are used in many medical journals to summarize key results and how those results were produced. These abstracts have the benefit of being quick to read while increasing the retention of information, largely thanks to the use of structure to guide authors in precisely summarizing their work.

- Papers could provide a table summarizing their contributions and the important design or implementation details that reflect the contribution.

All of these approaches provide the reader with data missing in *ad hoc* arguments: a specific map from the claims made about a sandbox to evidence that justifies the claim

has been met. They are also necessarily qualitative, but as we saw earlier, arguments are often used where more rigorous approaches are currently intractable. We believe that adding structure to these arguments is a reasonable advancement of the state of practice in sandbox validation.

## Sandbox and policy usability

Sandbox and policy usability are concerns of interest to the following stakeholders: *practitioners* that must correctly use sandboxes to improve the security postures of their systems and *users* that must work with sandboxed applications. Some security researchers do attempt to make their sandboxes more usable by providing policy management or reducing requirements on the user, but usability is definitely not a focus of any of the papers in our sample.

Our data shows that, with very few exceptions, sandbox researchers thoroughly evaluate the performance of their sandboxes. Why is there focus on this practical concern but not on usability? We observe that a focus on performance evaluation is partially motivated by the fact that overhead is relatively easy to quantify, but we also saw many cases where researchers were explicitly concerned with whether or not a sandbox was too resource intensive for adoption. The latter is a reasonable concern; *Szekeres et al. (2013)* pointed out that many mitigations for memory corruption vulnerabilities are not adopted because performance concerns outweigh protection merits.

While the idea that performance is an important adoption concern is compelling and likely reflects reality, we cannot correlate performance with the adoption of the sandboxes in our set. We cannot find a correlation because the sandboxes and their techniques in our set remain almost entirely unadopted. We only found four cases where sandboxes in our set were either directly adopted or where the techniques they evaluate are clearly implemented in a different but adopted sandbox. A lack of adoption is present even for techniques where performance and applicability have been improved over multiple decades (e.g., SFI). Three of the adopted sandboxes were created by the industry itself or by entities very closely tied to it: Google NaCl was designed with the intention of adopting it in Google Chrome in the short term (*Yee et al., 2009*; *Sehr et al., 2010*) and the paper on `systrace` was published with functioning open source implementations for most Unix-like operating systems (*Provos, 2003*). While the case for adoption is weaker, Cells (*Andrus et al., 2011*) is a more advanced design than one VMware developed in parallel (*Berlind, 2012*), although the sandboxes both aim to partition phones into isolated compartments using virtualization (e.g., one for work and one for personal use). More recently, Microsoft has stated that Visual Studio 2015 will ship with an exploit mitigation that we believe is equivalent to what the research community calls CFI (*Hogg, 2015*). A third party analysis supports this belief, however the uncovered implementation details differ from the techniques implemented in published research (*Tang, 2015*).

We argue that the need to evaluate the usability of our sandboxes is evidenced by the observation that performance and security evaluation are not sufficient to drive adoption. Usability is of particular concern in cases where the sandbox requires developers without

security expertise (1) to re-architect applications to apply the sandbox and/or (2) to develop a security policy. In practice, it is quite common for developers without a security focus to apply sandboxes, particularly Java's. In fact, usability issues have factored into widely publicized vulnerabilities in how sandboxes were applied to Google Chrome and Adobe Reader as well as the many vulnerable applications of the Java sandbox (*Coker et al., 2015*). In all of these cases applying the sandbox is a relatively manual process where it is difficult for the applier to be sure he is fully imposing the desired policy and without missing relevant attack surfaces. These usability issues have caused vulnerabilities that have been widely exploited to bypass the sandboxes. We call on the community to evaluate the following usability aspects of their sandboxes where appropriate:

- The intended users are capable of writing policies for the component(s) to be sandboxed that are neither over- or under-privileged.
- Policy enforcement mechanisms can be applied without missing attack surfaces that compromise the sandbox in the targeted component(s).
- Source code transformations (e.g., code re-writing or annotations) do not substantially burden future development or maintenance.
- The sandbox, when applied to a component, does not substantially alter a typical user's interactions with the sandboxed component.

Ideally many of these points would be evaluated during user studies with actual stakeholders. However, we believe that we can make progress on all of these points without the overhead of a full user study, particularly because we are starting from a state where no usability evaluations are performed. For example, authors can describe correct ways to determine what privileges in their policy language a component needs or even provide tools to generate policies to mitigate the risks presented by under- and over-privileged policies. Similarly, tooling can be provided to help users install policy enforcement mechanisms or check that manual applications of a mechanism are correct. Sandbox developers can transform or annotate representative open source applications and use repository mining (http://msrconf.org) to determine how sandbox alternations are affected by code evolution present in the repository (*Kagdi, Collard & Maletic, 2007*; *Yan, Menarini & Griswold, 2014*; *Mauczka et al., 2010*; *Stuckman & Purtilo, 2014*). Finally, a summary of how the sandbox qualitatively changes a user's experience with a sandboxed component would provide a gauge for how much the sandbox burdens end-users.

## ENABLING META-ANALYSIS

We believe a key contribution of this work is the use of multi-disciplinary and systematic methodologies for drawing conclusions about a large body of security techniques. In this section, we discuss the generalizability of our methodology and suggest other areas to which it can be applied. Then, we discuss some challenges that we faced when doing this research and suggest changes that would address these challenges.

## Generalizability of methodology

The methodology employed in this paper is based on two research approaches: qualitative Content Analysis and Systematic Literature Reviews. Qualitative Content Analysis is primarily used in the humanities and social sciences. Systematic Literature Reviews were first applied to medical studies and are used primarily in empirical fields. The differences between sandboxing papers are bigger than the differences between studies of a particular cancer treatment. In addition, sandboxing papers do not fit into the "native" domains of either approach—their primary contributions are designs, techniques, and implementations.

The result of these differences is that most literature reviews and systemizations in computing are done in an ad hoc manner. Our computing research is worthy of a more rigorous approach and we think the methodology applied in this paper can and should be applied to other topics. In fact, any topic of active research where the primary contributions is an engineered artifact, but without a clear and precise definition, would be amenable to our approach. These topics span computing research from software engineering (e.g., service oriented architecture, concurrent computation models) to systems (e.g., green computing, no instruction set computing) to human–computer interaction (e.g., GUI toolkits, warning science).

## Meta-analysis challenges and suggested solutions

In our experience, the biggest roadblock standing in the way of applying the same techniques to other segments of the research community lies in the difficulty involved in collecting analyzable metadata about papers. We experienced several fixable issues:

- The major publishers in computer science—IEEE, ACM, and Usenix—do not provide publicly available mechanisms to collect metadata and either rate limit or outright ban scraping. [9] In our case, the painstaking process of collecting and curating analyzable metadata across several sources limited our ability to explore hypotheses about our dataset's papers and their relationships to publications not in the set.

- The metadata is limited and contains little semantic content—typically the metadata includes the authors, title, data, and DOI, but little else. If abstracts and keywords were easier to harvest we could have more systematically derived topics of interest within the sandboxing community.

- Links to papers on publisher websites use internal identifiers (e.g., http://dl.acm.org/citation.cfm?id=2498101) instead of DOI. This makes it difficult to reference papers across publisher repositories.

- Conference websites have inconsistent layouts, which increases the difficulty of data collection.

We believe easier access to this data would have allowed us to draw more conclusions about how sandboxing papers are related and how the sandboxing landscape has evolved over time. For example, we explored the idea of using a more developed citation graph

[9] In at least one case ACM provided a copy of their digital library for scraping (*Bergmark, Phempoonpanich & Zhao, 2001*)

than Fig. 2 to trace the lineage of sandboxing techniques, but found the required resource expenditures were outside of our means. This data may provide support for explanations regarding the lack of advancement in security validation practices (e.g., by showing an emphasis on a different but important dimension of advancement). These points are important to understand how we got to the current state of practice, thus improving our ability to recognize and advance means for enhancing our results.

On another data collection point, we averaged about 45 min per paper to code the data necessary to answer our research questions. While we do not claim that our research questions are of universal interest to the sandboxing community, we did observe that papers that answer all or most of the questions in the abstract are often clearly written throughout and easy to interpret. A small minority of sandboxing papers have far less specific abstracts. In these cases, the papers often took double the average time to comprehend and interpret. It may be useful to strive to clearly answer questions like ours in future papers to show practitioners the value sandbox researchers bring to the table.

## THREATS TO VALIDITY

Due to the complexity of the text and concepts we are interpreting, there is some risk that other coders would assign quotes to different codes. Different codes will change the results, but we believe this risk is mitigated through our tests of the coding frame and by our efforts to select clear quotes. Furthermore, the correlative nature of our results ensures that a few code divergences will not dramatically change the analysis's outcomes.

The primary risk is that we are missing relevant quotes that add codes to our dataset. This is typically mitigated in QCA by fully segmenting the text, but we decided against that strategy because of the very large data set we studied and irrelevance of most of the text to our goals. We did search PDFs for relevant keywords we observed were commonly linked to specific codes throughout the process (e.g., "proof", "available" to find the availability of sandbox artifacts for evaluation, "experiment" to signal a case study or benchmark, etc.) to decrease the odds of missing a code. While this does mitigate the risk, it is still likely that our results under-approximate the state of the sandboxing landscape.

## CONCLUSION

We systematically analyzed the sandboxing landscape as it is represented by five top-tier security and systems conferences. Our analysis followed a multidisciplinary strategy that allowed us to draw conclusions backed by rigorous interpretations of qualitative data, statistics, and graph analysis. Based on our results, we conclude that the sandbox research community will benefit from the use of structured arguments in support of security claims and the validation of sandbox and policy usability. We suggested lightweight ways to move forward in achieving these goals. Our data also shows that there is a dearth of science regarding the management of security policies for sandboxes, although we did not discuss this gap in depth.

### Funding

This material is based upon work supported by the US Department of Defense through the Office of the Assistant Secretary of Defense for Research and Engineering (ASD(R&E)) under Contract HQ0034-13-D-0004 and the National Security Agency under Lablet Contract H98230-14-C-0140. Any opinions, findings, and conclusions or recommendations expressed in this material are those of the author(s) and do not necessarily reflect the views of ASD (R&E) or NSA. The funders had no role in study design, data collection and analysis, decision to publish, or preparation of the manuscript.

### Grant Disclosures

The following grant information was disclosed by the authors:
US Department of Defense: HQ0034-13-D-0004.
National Security Agency: H98230-14-C-0140.

### Competing Interests

The authors declare there are no competing interests.

### Author Contributions

- Michael Maass conceived and designed the experiments, performed the experiments, contributed reagents/materials/analysis tools, wrote the paper, prepared figures and/or tables, performed the computation work, reviewed drafts of the paper.
- Adam Sales analyzed the data, contributed reagents/materials/analysis tools, performed the computation work, reviewed drafts of the paper.
- Benjamin Chung analyzed the data, performed the computation work, reviewed drafts of the paper.
- Joshua Sunshine conceived and designed the experiments, performed the experiments, wrote the paper, reviewed drafts of the paper.

### Data Availability

We have supplied all our data in the Supplemental Dataset files.

### Supplemental Information

Supplemental information for this article can be found online at http://dx.doi.org/10.7717/peerj-cs.43#supplemental-information.

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
