# Peer review of "A systematic analysis of the science of sandboxing"

_PeerJ Computer Science, doi:10.7717/peerj-cs.43_

## Round 0.1 · original submission · Minor Revisions

Thank you very much for submitting your paper entitled "A systematic analysis of the science of sandboxing".

The referees have provided thoughtful reviews. The paper has the potential to make a valuable contribution, yet to do so will require some minor revisions. A few of the key points are:

-- you should clarify that your work focuses on sandboxes intended to encapsulate software through the constraints imposed by a security policy. Assuming that this is actually your intent, this will mitigate a comment on this topic made by the first reviewer, while addressing a suggestion of the second reviewer.

-- provide for better reproducibility of your results by storing your data in a more stable long term repository

-- ensure that all references, including those in footnotes, are useable an that the paper is self-contained.

-- clarify "usability". The generality of the term implies a large user population, whereas the sandboxes may be used by domain experts.

This is important work and we hope that you will address the items above and other comments and suggestions of the referees.

Reviewer 1 ·

Basic reporting

This paper conducts an analysis of individual sandbox mechanisms and reports on
different approaches to sandboxing and evaluates the design space and trade-offs
of these mechanisms. This survey looks at 5 major conferences and conducts the
meta study on the papers of the last couple of years.

Experimental design

Only a small subset of conferences were evaluated. For generality, other
(top) conferences like ASPLOS, ATC, OOPSLA, PLDI must be considered as well.
Also, the authors should look at VEE, a conference, co-located with ASPLOS that
focuses on virtualization and sandboxes with a long detailed history on
sandboxing work. The dataset therefore seems limited and selective (likely to
reasonably constrain work for the authors).

In 4.2 you explain the problems of a user when applying a sandbox. I missed a
discussion from the perspectives of a system administrator and a programmer.
Those should be more involved than a regular user and have additional domain
knowledge that would be interesting.

Validity of the findings

The authors selected papers by reading through titles. Why not consider keywords
and abstract as well? Especially, in systems-related works. Also, a fulltext
search for sandbox should have been considered. Therefore, I find that the
search for related work is limited.

Additional comments

The authors use a large amount of footnotes, many of them containing shortened
links. The paper should be self-contained which it is not. Also, as a reviewer I
cannot follow shortened goo.gl links because they would leak my anonymity. I was
therefore unable to evaluate all this additional material that was referenced.

Generally, the paper is very well structured, clear, and well written.

Nit: line 359: you cannot say that you don't look at these conferences and in
the next line that you are not aware of any such paper. If you are not
evaluating these conferences then you are naturally not aware of the existence
of such a paper.

In Figure 2: names of authors, names of techniques, and names of mechanisms are
mixed together. Can you make this figure more consistent?

In Figures 3 and 4 you introduce a lot of metrics

Figures 5 and 6 are very similar (yet 5 is in much worse quality).

Typo, line 312: is anywhere the application is: remove first is
Typo, line 328: diversity to binaries to: diversity in binaries

·

Basic reporting

The article is written very clearly, following a logical structure. Sufficient motivation for the study is provided in the introduction, with the subsequent material solidly delivering on the promises made.
Based on the insights gained from the analysis, the authors propose a concise and comprehensive definition of the term "sandbox", report that the evaluation of sandboxes can be improved, and propose the use of systematic sandbox evaluation techniques as well as work on the usability front.

Figures are clean and well drawn. Figures 3 and 4 could be more usefully presented by combining them into a cluster column chart. The dendrograms can be improved by labelling their top branches, and by linking the leaves to the corresponding papers. Figure 7 can be improved by combining the clusters lying in the vertical axis into one cluster column chart and labelling the corresponding clusters. The meaning of Figure 7 should be expanded in the text.

Tables are used to good effect for the clear and succinct presentation of data. In particular, Table 6 provides an exemplary concise and informative overview of the surveyed work.

On lines 160-167 the authors describe how they selected the conferences based on quantitative study of the conferences where sandboxing papers were published. It would benefit the community and the paper's reproducibility to provide these data.

The observation that policy flexibility can be used as a usability bellwether is novel and very interesting.

The finding regarding a lack of benchmark suites to highlight a sandbox's vulnerabilities (line 476) could be added to the conclusions.

Experimental design

The article reports on the state of the art in sandboxing methods through the use of quantitative content analysis on related papers published in top security conferences. It is certainly within the journal's scope. The 13 research questions studied and the possible answers are clearly defined in Table 3. It is however unclear how the research questions were derived. The questions seem to place considerable weight on security policies, their usability, and evaluation, while e.g. placing less focus on other possible attributes, such as portability, security, or performance. The authors allude to this bias on lines 169-173 and 554-555. Focusing on a specific area is fine, as long as the authors explicitly clarify (and maybe also justify) the focus.

The choice of the conferences used can be also examined and justified by reference to the CORE Conference Ranking activity <http://www.core.edu.au/index.php/conference-rankings>.

The authors should be commended for their choice of quantitative content analysis as a method, which by itself is a valuable contribution to the field. The method is well described, though the reader must reference the supplementary data in order to understand its use. A few concrete examples could help the reader's understanding.

It would be nice to provide the rationale for the criteria used for picking papers (lines 180-190). Theoretical constructs for sandboxes could also be explicitly listed as outside the paper's scope (line 183).

The regression method used for adjusting citation ranks according to the publication year (line 289) should be explained.

Validity of the findings

The statistical techniques used are appropriate for answering the research questions, and are correctly used to support the conclusions.

The study's raw data are made available as Google documents accessible through Google shortened URLs. Given Google's relatively poor record in the long term support of its services <http://www.lemonde.fr/pixels/visuel/2015/03/06/google-memorial-le-petit-musee-des-projets-google-abandonnes_4588392_4408996.html>, I would recommend that the authors archive their data in a repository that explicitly supports the long-term preservation of scientific data.

Another reason for preferring static policies (line 317) could be that they could be more secure, because they cannot be modified into a less secure version.

It is unclear how the authors found that performance cannot be correlated with a sandbox's adoption (lines 470-474). Given the amount of work that is being done in the community to improve the performance of sandboxes, it is important to empirically substantiate the finding.

The discussion regarding the usability of sandoxes in section 5.2 is interesting and to the point. However, it is well known that usability is often at odds with security. Who would be the people targeted for the usability improvements of the aspects listed in lines 492-497? Given that these are probably experts, would they really benefit from the proposed usability improvements? Perhaps the term "usability" is misleading, and terms such as "expressiveness", "readability", "analyzability", and "maintainability" should be used instead.

The authors mention how lacking metadata in the bibliographic databases they used hindered their work. Could Scopus provide the authors with the data they required?

The conclusions section can be expanded to make justice to the considerable amount of work done by the authors, perhaps by summarizing their findings in actionable form.

Additional comments

The provided definition of a sandbox (line 137) can be improved by stating that it is an encapsulation mechanism *that is used to impose* a security policy on software components. This adds the use's intent, thus precluding technologies such as OS processes and virtual machines that are not explicitly used to sandbox components although they impose security policies.

On lines 117-118 the authors state that one might object to the use of "encapsulation, because it is typically used to refer to cases where the inside is protected from the outside, whereas sandboxes protect in the reverse direction. One can overcome this problem by considering that a sandbox encapsulates the external system (which needs to be protected) from the potentially malicious component.

Reviewer 3 ·

Basic reporting

This paper is a detailed survey and classification of sandboxing. The authors apply a systematic approach. This paper will help the community finding the most relevant material on sandboxing as well as how the different contributions compare.

Experimental design

The systematic methodology used to select and analyze papers on sandboxing includes steps like the selection of papers, the categorization of datasets, and the analysis of the datasets.

Validity of the findings

The results are based on statistical clustering of how the papers were coded, discovered trends in the dataset, and observations when reading the papers or reviewing the data. The results include a list of research questions stated in the papers, of policies, as well as the use of validation.
The results of this paper like Table 6 will allow the community to have a clear picture of the various contributions on sandboxing.
The authors included a section on the threats to validity.

Additional comments

This paper is well structured and written. This systematic analysis is a contribution to the field allowing the community to better compare published works. The analysis is carefully conducted.

---

## Round 0.2 · accepted · Accept

Thank you for this fine article. It will be a very useful contribution to the literature on sandboxes.

A few minor notes that you can fix at production:
- Line 180: should you use "led" instead of "lead"
- Line 332: should you use "exist" instead of "exists"

There are probably a few other minor nits, but these are the ones I caught.